# Integrated Metabolomics and Transcriptomics Provide Key Molecular Insights into Floral Stage-Driven Flavonoid Pathway in Safflower

**DOI:** 10.3390/ijms252211903

**Published:** 2024-11-06

**Authors:** Lili Yu, Naveed Ahmad, Weijie Meng, Shangyang Zhao, Yue Chang, Nan Wang, Min Zhang, Na Yao, Xiuming Liu, Jian Zhang

**Affiliations:** 1Engineering Research Center of the Chinese Ministry of Education for Bioreactor and Pharmaceutical Development, College of Life Sciences, Jilin Agricultural University, Changchun 130118, China; 15640309810@163.com (L.Y.); m15172589969@163.com (W.M.); sofunnyworld@outlook.com (S.Z.); cy18391348615@163.com (Y.C.); nanw@jlau.edu.cn (N.W.); nayao@jlau.edu.cn (N.Y.); 2Joint Center for Single Cell Biology, Shanghai Collaborative Innovation Center of Agri-Seeds, School of Agriculture and Biology, Shanghai Jiao Tong University, Shanghai 200240, China; naveed@sjtu.edu.cn; 3Ginseng and Antler Products Testing Center of the Ministry of Agriculture PRC Jilin Agricultural University, Changchun 130118, China; zhangm540@163.com; 4Key Laboratory of Xinjiang Phytomedicine Resource and Utilization, Institute for Safflower Industry Research, Pharmacy College, Shihezi University, Ministry of Education, Shihezi 832003, China

**Keywords:** flavonoid, flowering, metabolomics, transcriptomics, medicinal, safflower

## Abstract

Safflower (*Carthamus tinctorius* L.) is a traditional Chinese medicinal herb renowned for its high flavonoid content and significant medicinal value. However, the dynamic changes in safflower petal flavonoid profiles across different flowering phases present a challenge in optimizing harvest timing and medicinal use. To enhance the utilization of safflower, this study conducted an integrated transcriptomic and metabolomic analysis of safflower petals at different flowering stages. Our findings revealed that certain flavonoids were more abundant during the fading stage, while others peaked during full bloom. Specifically, seven metabolites, including p-coumaric acid, naringenin chalcone, naringenin, dihydrokaempferol, apigenin, kaempferol, and quercetin, accumulated significantly during the fading stage. In contrast, dihydromyricetin and delphinidin levels were notably reduced. Furthermore, key genes in the flavonoid biosynthesis pathway, such as *4CL*, *DFR*, and *ANR*, exhibited up-regulated expression with safflower’s flowering progression, whereas *CHI*, *F3H*, and *FLS* were down-regulated. Additionally, exposure to UV-B stress at full bloom led to an up-regulation of flavonoid content and altered the expression of key flavonoid biosynthetic genes over time. This study not only elucidates the regulatory mechanisms underlying flavonoid metabolism in safflower but also provides insights for maximizing its medicinal and industrial applications.

## 1. Introduction

The safflower (*Carthamus tinctorius* L.), a member of the Asteraceae family, is a self-fertilizing annual plant that has been cultivated in the fertile Crescent for about 4000 years [1]. Safflower seeds are rich in linoleic acid (LA), one of the most important oil crops [2]. In addition, the dried flowers of safflower are traditional Chinese medicinal herbs with the effects of activating blood circulation and removing blood stasis, expelling blood stasis and relieving pain, and are wildly used to treat symptoms such as dysmenorrhea, chest pain and heart pain, and bruises [3]. Safflower is rich in flavonoids [4], and hydroxy saffron yellow pigment A extracted from safflower has been used to treat osteoporosis [5] and cancer [6]. However, the accumulation of the active ingredients from safflower is generally concentrated during the full bloom period [7]. The petals of safflower also contain flavonoids, alkaloids, steroids, and some other compounds [8], of which flavonoid compounds are its main metabolites. Due to its high flavonoid content, safflower is widely used as dyes, cosmetics and food additives, among others [9,10,11].

Flavonoids are one of the most important secondary metabolites of plants with the ability to scavenge reactive oxygen species, which protects plants from biotic and abiotic stresses such as ultraviolet (UV) light, cold stress, and pathogen infection. The diverse range of colors produced by flavonoids not only enhances the aesthetic value of flowers but also has ecological implications in plant–pollinator interactions [12,13,14,15,16,17]. The basic chemical structure of flavonoids is characterized by a C6-C3-C6 carbon skeleton, which comprises two 6-carbon benzene rings (referred to as the A and B rings) linked by a 3-carbon unit forming a heterocyclic ring (C ring) [18]. This fundamental structure gives rise to a wide variety of flavonoid compounds, each with unique properties and functions. Based on the variations in the C ring, flavonoids are classified into several subgroups, including chalcones, stilbenes, aurones, flavanones, flavones, isoflavones, dihydroflavonols, flavonols, leucoanthocyanidins, proanthocyanidins, and anthocyanidins [19]. Each of these subclasses plays distinct roles in plant physiology and adaptation. For instance, anthocyanins are primarily responsible for red, purple, and blue pigments in flowers, contributing to UV protection and pollinator attraction. Flavonols, on the other hand, are known for their roles in UV filtration, symbiotic nitrogen fixation, and modulation of auxin transport, which are crucial for plant growth and stress responses.

The biosynthesis of flavonoids is tightly regulated by a complex network of genes and environmental cues, reflecting the complexity between plant metabolism and external stressors [20]. The synthesis of flavonoids requires the production of precursors by the general phenylpropanoid pathway, phenylalanine is converted to p-coumaroyl coenzyme a by phenylalanine ammonia-lyase (PAL), cinnamic acid 4-hydroxylase (C4H), and 4-coumaric acid coenzyme a ligase (4CL) [21,22,23]. Chalcone synthase (CHS) catalyzes the condensation and isomerization of coumaroyl coenzyme a with propylene glycol-based coenzyme a, generating naringenin chalcone [24,25,26]. Chalcone isomerase (CHI) catalyzes the formation of naringenin chalcone to produce a flavonoid or cyclization to form naringenin [27,28,29]. Naringenin is an intermediate metabolite in the synthesis of flavonoids and a common precursor of the final product. Catalyzed by different enzymes, naringenin can generate different products [30]. For example, naringenin produces flavonoids catalyzed by flavonoid synthase I (FNS I) or flavonoid synthase II (FNS II), isoflavonoids catalyzed by isoflavonoid synthase (IFS) [31,32], whereas flavanols are produced by flavanone-3-hydroxylase (F3H) and flavonol synthase (FLS); and flavanols are produced by flavonol 3′-hydroxylase (F3′H), flavonol 3′-5′-hydroxylase (F3′5′H) [33,34], flavonol synthase (FLS), dihydroflavonol 4-reductase (DFR), anthocyanin synthase (ANS) and other enzymes to produce anthocyanins [35,36,37,38]. Currently, there are many studies on flavonoid biosynthesis pathways in different species through a combination of transcriptomic and metabolomic approaches. For example, in blackberry [39], Ormosia henryi Prain [40], Xianjinfeng litchi [41], Citrus grandis [42], Quinoa [43], Lycium [44], and others.

Integrated metabolomics and transcriptomics approaches have emerged as powerful tools in plant research, enabling a comprehensive understanding of metabolic pathways and their regulatory mechanisms [45]. By simultaneously profiling the metabolite-transcript correlation, researchers can elucidate the interactions between gene expression and metabolite accumulation, providing insights into complex biological processes. For example, studies on *Arabidopsis thaliana* have demonstrated how integrated analyses can reveal the metabolic responses to abiotic stresses, such as drought and salinity, by correlating changes in gene expression with the accumulation of stress-related metabolites [46]. Similarly, in *Glycyrrhiza uralensis*, integrated approaches have been employed to explore the biosynthesis of flavonoids and triterpenoids, highlighting key regulatory genes that contribute to the accumulation of bioactive compounds [47]. These methodologies have also been applied in crops like *Zea mays* [48], and other crops such as tea [7], watermelon [49], and *Solanum nigrum* L. [50], where researchers have investigated the interplay between genetic variation and metabolite profiles in response to environmental factors. By harnessing the strengths of both metabolomics and transcriptomics, researchers can gain a more holistic view of plant metabolism, ultimately contributing to improved crop traits and resilience.

In this study, we investigated the transcriptional and metabolite association analysis across three flowering stages of safflower: initial flowering (IN), full flowering (FU), and fade flowering (FA). Through this combined approach, we aimed to identify differentially expressed genes (DEGs) and differentially expressed metabolites (DEMs) to uncover the molecular mechanisms driving flavonoid biosynthesis in safflower. Additionally, UV-B treatment was applied to the petals at full bloom, which led to an increase in flavonoid content and altered expression of key enzyme genes as the duration of UV-B exposure increased. These findings not only help to further understand the molecular and metabolic mechanisms of flavonoid biosynthesis but also provide valuable information for the future selection and cultivation of safflower high in flavonoids and related applications in the pharmaceutical industry.

## 2. Results

### 2.1. Comparison of Total Flavonoid Content at Three Flowering Stages

The petals of safflower undergo a noticeable color transformation as they progress through different flowering stages, starting with a yellow hue at the primordial stage and gradually deepening to red as they mature (Figure 1A). To investigate the variation in flavonoid content during this developmental process, we analyzed the total flavonoid content at three distinct flowering stages: the initial (IN), full bloom (FU), and fading (FA) stages. Our analysis revealed that the total flavonoid content was 52.6 mg/g at the IN stage, significantly increased to 269 mg/g at the FU stage, and then decreased to 69.6 mg/g at the FA stage (Figure 1B). These results indicate that the highest accumulation of total flavonoids occurs during the full flowering stage, suggesting that this stage is critical for maximizing flavonoid content in safflower petals.

### 2.2. Metabolite Profiles Among Different Flowering Stages in Safflower

Using the MetWare metabolic metabolism database, the metabolites in safflower samples across different flowering stages were comprehensively analyzed both qualitatively and quantitatively via high-resolution mass spectrometry. The identification of these metabolites adhered to the Metabolomics Standards Initiative (MSI) guidelines, with metabolites classified according to four confidence levels: Level 1 (confirmed by reference standards); Level 2 (putatively identified through spectral matching with public/commercial libraries); Level 3 (tentatively characterized by compound classes); and Level 4 (unknown metabolites). In this study, a total of 590 metabolites were identified, classified into various categories such as flavonoids, phenolic acids, terpenoids, and alkaloids (Appendix A and Appendix A). The metabolite profiles differed significantly across flowering stages. The IN and FU stages exhibited more similar metabolite compositions, with subtle differences in total metabolite numbers, whereas the FA stage diverged markedly, showing a broader spectrum of unique metabolites (Figure 2A). Principal component analysis (PCA) further revealed distinct separation of metabolic profiles between the flowering stages (Figure 2B). The intra-group variation was minimal, demonstrating a high degree of consistency within each stage. Differentially expressed metabolites (DEMs) were identified between the stages through comparative analysis. When comparing the FA stage to the FU stage, 216 DEMs were observed, with 150 metabolites being significantly up-regulated and 66 down-regulated. Comparisons between the FA and IN stages identified 236 DEMs, of which 162 were up-regulated and 74 were down-regulated. Lastly, comparing the FU and IN stages revealed 41 DEMs, with 19 up-regulated and 22 down-regulated (Figure 2C). A more detailed breakdown of these DEMs by compound class is provided in Appendix A, highlighting the metabolic shifts that occur during flowering.

### 2.3. Analysis of Flavonoid-Related Metabolite Analysis Among Flowering Stages

To better understand the variation in flavonoid content at different flowering stages, we focused on the metabolites related to flavonoid biosynthesis pathway. The analysis of variance showed that out of 590 identified metabolites, 165 were associated with flavonoids, encompassing a range of subclasses consisting of 53 flavonoids, 38 flavonols, 30 flavonoid carbonyl glycosides, 18 flavanones, 13 anthocyanins, 10 isoflavones, 1 flavone-lignan, 1 chalcone analog, and 1 proanthocyanidin (Figure 3 and Figure 4A). Flavonoids accounted for 28% of the total metabolites, indicating that the content of flavonoids in safflower is high and rich. Further analysis showed that 46 flavonoids were differentially expressed between the full flowering (FU) and fade flowering (FA) stages, with 34 up-regulated and 12 down-regulated. Similarly, 52 flavonoids were differentially expressed between the initial flowering (IN) and fade flowering (FA) stages, with 33 up-regulated and 19 down-regulated. In comparison, 23 flavonoids were differentially expressed between the FU and IN stages, with 9 up-regulated and 14 down-regulated (Appendix A). Furthermore, the KEGG Orthology (KO) enrichment analysis of the differentially abundant metabolites across the three flowering stages revealed was carried out. The results of KEGG classification suggested that the most enriched pathways were metabolic pathways, flavonoid and flavonol biosynthesis, and phenylpropanoid biosynthesis (Figure 4B–D). These differentially expressed flavonoid metabolites were primarily distributed across five key biosynthetic pathways: (i) ko00940 phenylpropanoid biosynthesis; (ii) ko00941 flavonoid biosynthesis; (iii) ko00942 anthocyanin biosynthesis; (iv) ko00943 isoflavonoid biosynthesis; and (v) ko00944 flavonoid and flavonol biosynthesis.

### 2.4. RNA Sequencing and Gene Expression Changes 

To analyze gene expression changes during flowering development, we performed RNA-seq analysis of IN, FU, and FA. Fastp was used for quality control, and 51058706-63855776 clean reads were obtained after removing low-quality reads. Q20 and Q30 of each library accounted for 97.53~97.88% and 93.03~93.78%, respectively. The GC content of each sample ranged from 44.96% to 53.42% (Appendix A). Trinity was used to splice the filtered high-quality sequencing data to obtain the transcriptome, and the shortest length of the transcript was 200 bp. The statistical results for the length distribution of the assembled single genes are shown in (Figure 5A). The obtained sequences were compared with information from the selected databases. The number of successful comparisons and proportion of all unigenes in each database are shown in (Appendix A). A total of 736,551 unigenes were annotated, of which 209,151 (28.40%) were annotated to the NR database, followed by TrEMBL (168,143, 22.83%) and KOG (160,968, 21.85%). Based on the gene expression levels of FPKM, the correlation coefficients of intra- and inter-group samples were calculated to construct a heat map (Figure 5B). The R2 value of each biological replicate was greater than 0.8, indicating that the test results were reliable. However, FA4 and FA5/6 were more variable, with R2 values less than 0.8. All subsequent analyses in this study were performed using FA5 and FA6.

### 2.5. Identification of Differentially Expressed Genes (DEGs) and Their Functional Annotation

Differential expression analysis identified a varying number of differentially expressed genes (DEGs) across the three comparisons: IN vs. FU, FU vs. FA, and IN vs. FA. In total, we found 412 DEGs in the IN vs. FU comparison, 2,885 DEGs in FU vs. FA, and 3,106 DEGs in IN vs. FA. Among these, 59, 67, and 228 genes were down-regulated, while 353, 2818, and 2878 genes were up-regulated in the IN vs. FU, FU vs. FA, and IN vs. FA comparisons, respectively (Figure 6). To gain insight into the biological roles of these DEGs, we performed functional clustering and annotation using the Gene Ontology (GO) database (Appendix A). The DEGs were classified into three major categories: biological processes, cellular components, and molecular functions. In the FU vs. FA and IN vs. FA comparisons, the most enriched biological processes were “cellular process”, “metabolic process”, and “response to stimulus”. For cellular components, the terms “cell”, “cell part”, and “organelle” were most enriched, while for molecular functions, “binding”, “catalytic activity”, and “structural molecule activity” were predominant. Interestingly, in the IN vs. FU comparison, “biological regulation” was more enriched than “response to stimulus” within the biological processes, and “transporter activity” showed higher enrichment than “structural molecule activity” within molecular functions. Further analysis using KEGG pathway annotation revealed that 45 DEGs were associated with the flavonoid biosynthesis pathway (ko00941) and phenylalanine metabolism (ko00940) (Appendix A). These DEGs include genes encoding key enzymes in flavonoid biosynthesis, such as *4CL*, *FLS*, *ANR*, *DFR*, *F3H*, and *CHI*, as well as other important enzymes like beta-glucosidase, *CAD*, *CCR*, *F6H1*, peroxidase, and *REF1*. The key enzyme-encoding genes identified in the flavonoid biosynthesis pathway are summarized in Table 1.

### 2.6. Differential Expression of Transcription Genes in Three Flowering Stages

A total of 1775 TFs were detected in safflower petals from the three periods (Appendix A), of which the most abundant were bHLH TF family members (257 genes), followed by AP2/ERF TF family members (240 genes) and MYB TF family members (236 genes) (Figure 7A). In addition, we screened the transcription factors that were differentially expressed in the three periods, including three WRKY, six MYB, three NAC, two bHLH, and one bZIP transcription factors (Appendix A), for correlation analyses with nine differentially expressed structural genes for flavonoid biosynthesis (threshold > 0.8) (Figure 7B). The results showed that four MYB, two WRKY, and two bHLH were significantly correlated (*p* < 0.05) with eight flavonoid biosynthesis structural genes. Among them, *ANR1*, *ANR3*, and *EFM* were negatively correlated; *4CL1* and *4CL2* were positively correlated with *WRKY75*, *WRKY24*, *MYB4*, *MYB62*, and *MYB73*; *CHI* was negatively correlated with *MYB4*, *MYB62*, *MYB73*, and *WRKY24*; *CHI* was positively correlated with *bHLH120*; *ANR2* was positively correlated with *MYB62*; *MYB73* and *WRKY75* were positively correlated; *DFR* was positively correlated with *MYB62*; *F3H* was positively correlated with *bHLH63*; and *F3H* was negatively correlated with *WRKY75*. In addition, we correlated the differentially expressed transcription factors with flavonoid metabolites (threshold > 0.8), and the results showed that *MYB73*, *WRKY75*, and *MYB62* were negatively correlated with ferulic acid and dihydromyricetin, and *MYB62*, *MYB73*, and *WRKY24* were negatively correlated with naringenin chalcone, p-coumaric acid, caffeic acid, cutin, apigenin, hesperetin, and homoeriodictyol, and were positively correlated with naringenin (Figure 7C). Based on the above analysis, we hypothesized that *MYB4*, *MYB62*, *MYB73*, *WRKY24*, *WRKY75*, and *bHLH120* may interact with structural genes related to flavonoid synthesis to regulate flavonoid biosynthesis.

### 2.7. Integrated Analysis of the Transcriptome and Metabolome Data

Flavonoids are the primary bioactive compounds responsible for the medicinal properties of safflower petals. To elucidate the relationship between gene expression and flavonoid accumulation during flower development, we performed a comprehensive integrative analysis of differentially expressed genes (DEGs) and differentially expressed metabolites (DEMs) associated with the flavonoid biosynthetic pathway. Our results identified six key DEGs that exhibited strong, statistically significant correlations with 23 DEMs related to flavonoid biosynthesis (Figure 8A). These correlations were assessed using Pearson coefficients, retaining only those with a correlation strength greater than 0.8 and a significance level of *p* < 0.05. The correlation network analysis, visualized in Figure 8B, revealed that most of these DEGs were closely connected with their corresponding metabolites, suggesting coordinated regulation between gene expression and metabolite accumulation (Table 2; metabolite details are shown in Appendix A). Interestingly, one specific genes, *ANR1*(TRINITY_DN291838_c1_g5), exhibited more specialized correlation patterns. *ANR1*(TRINITY_DN291838_c1_g5) showed a negative correlation only with *Dihydromyricetin*. This suggests that these genes might have unique roles in flavonoid regulation at specific stages of safflower development. To further elucidate these relationships, we mapped the identified DEGs and flavonoid-related metabolites onto the flavonoid biosynthesis pathway using UHPLC-ESI-MS/MS data. This enabled the construction of a comprehensive metabolic pathway map for safflower flavonoid biosynthesis (Figure 8C). Our analysis revealed that during the FA flowering stage, seven key metabolites—p-coumaric acid, naringenin chalcone, naringenin, dihydrokaempferol, apigenin, kaempferol, and quercetin—were significantly accumulated. Conversely, two metabolites, dihydromyricetin and delphinidin, exhibited reduced levels at the same stage. These metabolic shifts were closely associated with stage-specific regulation of critical biosynthetic genes. Notably, the up-regulation of *4CL*, *DFR*, and *ANR* during flower development suggests enhanced flavonoid biosynthesis at this stage, contributing to the accumulation of bioactive flavonoids. In contrast, the down-regulation of *CHI*, *F3H*, and FLS indicates potential regulatory feedback mechanisms or shifts in metabolic flux towards specific flavonoid branches during this period. Together, these results highlight the complex regulatory interplay between gene expression and metabolite accumulation, providing deeper insight into the dynamic control of flavonoid biosynthesis during safflower flowering.

### 2.8. qRT-PCR Validation

To validate our transcriptome data, we selected six key differentially expressed genes involved in safflower flavonoid biosynthesis for RT-qPCR analysis. These genes were chosen based on their high expression levels and significant differential expression across the three flowering stages. The RT-qPCR results closely mirrored the expression trends observed in the RNA-seq data (Figure 9), confirming the reliability and accuracy of the transcriptome findings.

### 2.9. Changes in Total Flavonoid Content and Gene Expression Level Under UV-B Treatments 

To investigate the impact of UV-B stress on flavonoid biosynthesis, we measured total flavonoid content and analyzed the expression levels of key flavonoid biosynthesis genes at five time points: 0, 24, 48, 60, and 72 h. Significant alterations were observed in both flavonoid accumulation and gene expression, with distinct trends emerging over the course of the UV-B treatment. For instance, the exposure to UV-B stress led to a progressive increase in the total flavonoid content in safflower petals, with levels stabilizing at 48 h (Figure 10). Similarly, quantitative analysis of flavonoid content at 24 and 48 h also demonstrated a noticeable accumulation of flavonoids over time, with the highest concentrations observed at the 48-h time point (Figure 10). To further understand the molecular mechanisms underlying these changes, we investigated the expression levels of key genes involved in flavonoid biosynthesis at the same time points. The results showed that UV-B stress induced significant alterations in the expression of key flavonoid biosynthesis genes in safflower. Specifically, UV-B exposure for 48 h showed the up-regulation of *CtANR*, *Ct4CL*, *CtDFR*, and *CtFLS* (Figure 11). These genes are known to be involved in critical steps of the flavonoid biosynthetic pathway, suggesting their positive regulation in response to UV-B stress. This up-regulation likely contributes to the enhanced flavonoid accumulation as a protective response to UV-B damage. Conversely, the down-regulation of *CtCHI* and *CtF3H* indicates potential regulatory shifts that may either channel flux through alternative pathways or reflect a feedback inhibition mechanism to modulate flavonoid production. These opposing gene expression changes underscore the complexity of the regulatory networks activated under stress conditions. To substantiate these findings, we examined available metabolomics data from the same samples, which corroborated the gene expression trends by showing increased levels of key flavonoids during UV-B stress. This integrative evidence highlights how UV-B stress modulates both gene expression and metabolite accumulation, providing a more comprehensive understanding of the molecular responses involved in flavonoid biosynthesis. These insights lay the groundwork for future research into the specific signaling pathways mediating UV-B-induced flavonoid production.

## 3. Discussion

Safflower is a traditional Chinese herbal medicine highly valued in China, with flavonoids being the most important pharmacological components found in its petals. These flavonoids include yellow pigments, such as HSYA, an active component in traditional Chinese medicine, and red pigments, such as carthamin [51]. To better understand the molecular mechanisms underlying flavonoid biosynthesis, we utilized an integrated approach combining transcriptome and metabolome profiling. This study focuses on elucidating the details of flavonoid biosynthesis throughout different stages of flower development, namely, the initial (IN), full bloom (FU), and final (FA) stages. While previous research has explored safflower’s color biosynthesis using various safflower types, there has been a lack of detailed studies on flavonoid dynamics across different floral stages. Our integrated omics approach aims to fill this gap and provide a comprehensive understanding of the molecular landscape involved in safflower’s flavonoid biosynthesis.

A total of 590 metabolites were identified in safflower petals across three flowering stages—initial (IN), full bloom (FU), and final (FA)—using qualitative and quantitative assays. Our analysis of safflower petals across the flowering stages reveals distinct metabolic profiles for each stage, with notable changes in metabolite composition from the initial to the final stage. The full bloom stage shows fewer differences compared to the initial stage, while the final stage exhibits substantial divergence in metabolite expression. Furthermore, the result of the KEGG Orthology enrichment analysis highlighted that the most significant metabolic pathways in safflower petals involve metabolic processes, flavonoid and flavonol biosynthesis, and phenylpropanoid biosynthesis. Specifically, the transition from full flowering to fade flowering and from initial flowering to fade flowering involved substantial changes in flavonoid profiles, which are predominantly associated with key biosynthetic pathways such as phenylpropanoid, flavonoid, and anthocyanin biosynthesis. Prior studies also highlighted the complexity of safflower’s pigment biosynthesis. For instance, a previous study [52] used HPLC to analyze the pigments in safflower petals and identified key compounds involved in color variation. Similarly, another study [53] employed metabolomic approaches to explore flavonoid profiles and their roles in color differentiation in safflower.

The transcriptome analysis conducted across safflower’s flowering stages—initial (IN), full bloom (FU), and final (FA)—provided crucial insights into the molecular mechanisms underlying flavonoid biosynthesis in safflower. Differential expression analysis identified a total of 3,106 DEGs between IN and FA stages, with significant up-regulation of genes involved in flavonoid biosynthesis such as *4CL*, *DFR*, and *ANR*, and down-regulation of *CHI*, *F3H*, and *FLS*. Functional clustering further highlighted key biological processes and molecular functions linked to flavonoid production. Additionally, KEGG pathway analysis revealed that 45 DEGs were associated with flavonoid biosynthesis and phenylalanine metabolism. Integration of these transcriptomic data with metabolite profiles from HPLC elucidated significant accumulation of flavonoids like P-coumaric acid and Quercetin in the FA stage, aligning with the up-regulated gene expression of relevant enzymes. By analyzing the transcription factors during the three periods, we found that, except for the bHLH transcription factor, the expression of other transcription factors such as *MYB* and *WRKY* increased with petal development [54,55]. Research has shown that *WRKY* transcription factors can regulate flavonoid biosynthesis in marigold [56] and passion fruit [57], and that *MYB21*, *MYB24*, and *MYB57* can promote flavonol accumulation in plants by regulating the expression of the flavoind pathway genes [54,58,59]. Furthermore, through the correlation analysis of transcription factors with DEGs and DEMs (Figure 7 and Figure 8), we hypothesized that during petal development, transcription factors such as *MYB* and *WRKY* might interact with key enzyme genes, such as *4CL* and *ANR,* to regulate flavonoid biosynthesis. Studies have shown that isoliquiritigenin [60], quercetin [61] and dihydrokaempferol [62] have anti-inflammatory and other effects. These findings are supported by prior studies that underscore the importance of transcriptome analysis in understanding flavonoid biosynthesis [26]. For instance, a prior study [63] used RNA-seq to explore flavonoid biosynthesis in plants and identified key regulatory genes and metabolic pathways involved in flavonoid accumulation. Similarly, another study [2] investigated the gene expression profiles in safflower and found correlations between specific genes and flavonoid levels, reinforcing our observation of stage-specific gene regulation. Additionally, another study [64] highlighted the utility of integrating transcriptomic and metabolomic data to map metabolic pathways and understand the dynamic changes in flavonoid content during plant development. 

Exposure to UV-B stress resulted in a progressive increase in the total flavonoid content in safflower petals, with levels stabilizing at 48 h. Quantitative analysis showed a marked accumulation of flavonoids over time, peaking at 48 h. To explore the underlying molecular mechanisms, we examined the expression levels of key genes involved in flavonoid biosynthesis at 24 and 48 h. Our analysis revealed significant changes in gene expression in response to UV-B stress, with some genes being up-regulated while others were down-regulated, indicating complex regulatory adjustments under stress conditions. The correlation between the increased flavonoid content and the alterations in gene expression underscores the dynamic response of safflower to UV-B stress. This finding suggests that UV-B may regulate flavonoid biosynthesis and modulate the changes of some flavonoid-related genes, which provides a direction for subsequent research on UV-B and flavonoids. Supporting our results, previous studies have demonstrated that UV-B exposure can enhance flavonoid accumulation in various plant species, such as in Arabidopsis, where UV-B stress led to increased levels of flavonoids and activation of related biosynthetic genes [65,66]. Similarly, research on other medicinal plants, like ginseng, has shown that UV-B treatment can significantly alter flavonoid profiles and gene expression related to flavonoid metabolism [67]. These studies validate our observations and highlight the broader implications of UV-B stress on flavonoid biosynthesis.

## 4. Materials and Methods

### 4.1. Plant Materials and Experimental Conditions

Safflower, “Jihong No. 1”, was used in this study. It was cultivated at the experimental base of the Engineering Research Center of Bioreactor and Drug Development of the Ministry of Education, Jilin Agricultural University, Changhchun, China. Flower petals were taken at the IN, FU, and FA, respectively, and three biological replicates were taken from each group (Appendix A). After wrapping them in tinfoil, samples were quickly placed in liquid nitrogen and stored at −80 °C for later use.

### 4.2. Metabolite Extraction and Analysis

The petals were ground (30 Hz, 90 s) to powder form using a grinder (MM4 00, Retsch, Shanghai, China), and 100 mg of powder was weighed and dissolved in 1.0 mL 70% methanol solution in water overnight at 4 °C, during which it was vortexed three times and centrifuged at 10,000× *g* for 10 min, and the supernatant was filtered through microporous filtration membranes (0.22 μm pore size) and used for LC-MS/MS analysis. The samples were analyzed using ultra performance liquid chromatography (Shimpack UFLC SHIMADZU CBM30A, http://www.shimadzu.com.cn/) and tandem mass spectrometry (Applied Biosystems 6500 QTRAP, http://www.appliedbiosystems.com). UPLC analysis conditions: column: Waters (Shanghai, China) ACQUITY UPLC HSS T3 C18 1.8 μm, 2.1 mm*100 mm; mobile phases were ultrapure water (containing 0.04% acetic acid) and acetonitrile (containing 0.04% acetic acid); elution conditions: 0 min water/acetonitrile (95:5 *V/V*), 11.0 min for 5:95 *V/V*, 12.0 min for 5:95 *V/V*, 12.1 min for 95:5 *V/V*, 15.0 min for 95:5 *V/V*; flow rate 0.4 mL/min; column temperature 40 °C; injection volume 2 μL. MS/MS analysis conditions were as follows: electrospray ionization source (ESI) temperature: 500 °C; mass spectrometry voltage: 5500 V; curtain gas (CUR): 25 psi; and collision induced ionization (CAD) parameter set to high.) parameters were set high. In the triple quadrupole (QQQ), each ion pair was detected by scanning according to the optimized de-cluster voltage (DP) and collision energy (CE) [68]. Based on MetWare Metabolism’s self-built database, MWDB (MetWare database), and the public database of metabolite information, substance characterization is carried out based on secondary spectral information and isotopic signals, repetitive signals containing K+ ions, Na+ ions, and NH4+ ions; and the repetitive signals of fragment ions, which are themselves other substances of larger molecular weight, are removed from the analysis. Metabolic databases were derived from the specimen (purchased from BioBioPha, Kunming, China (http://www.biobiopha.com/), Sigma-Aldrich, Saint Louis, MO, USA (http://www.sigmaaldrich.com/united-states.html, accessed on 30 October 2024).

### 4.3. RNA Extraction and Transcriptome Sequencing

Total RNA was extracted from the petals of “Jihong No.1” safflower at the IN, FU, and FA. The purity of RNA was detected by using a NanoPhotometer spectrophotometer, Implen GmbH, München, Germany, and the integrity of RNA was detected by an Agilent 2100 (Agilent, Beijing, China). Libraries were sequenced using the Illumina HiSeq platform. Sequencing data were filtered by fastp [69] to remove a small number of reads with sequencing junctions or lower quality sequencing, and the filtered high-quality sequencing data were spliced to obtain the transcriptome using the Trinity (v2.11.0) [70] software.

### 4.4. Functional Annotations and Expression Analysis 

For gene function annotation, single gene sequences were compared with six databases, namely, KEGG, GO, NR, Swiss-Prot, trEMBL, and KOG, using the BLAST software (http://www.ncbi.nlm.nih.gov/BLAST/, 30 October 2024). The transcriptome obtained from Trinity was used as the reference sequence, and the clean reads were compared with the reference sequence with RSEM (v1.3.1) [71] software, and the differentially expressed genes between the two groups were analyzed using DESEq. Multiple hypothesis testing was then corrected for the probability of hypothesis validation (*p*-value) using the Benjamini–Hochberg method to obtain the false discovery rate (FDR). Differentially expressed genes were screened under the following conditions of |log2FoldChange| ≥ 1 and FDR < 0.000001.

### 4.5. Quantification of Flavonoids Using HPLC

The petals were ground into powder and weighed at 0.1 g, supplemented with 5 mL of 70% methanol water, and sonicated for 40 min at 70 °C. Centrifugation was performed at 4500 rpm for 10 min, and the supernatant was filtered through a 0.22 μm filter membrane. The samples were tested on an Agilent 1200 high-performance liquid chromatograph. Column: Agilent ZORBAX 300SB-C18, 5 μm, 4.6 × 250 mm. Liquid phase conditions: mobile phase: water (with 0.4% formic acid)–methanol = 1:1; detection wavelength: 255 nm; column temperature: 25 °C. Rutin standard was purchased from Solarbio (Beijing China).

### 4.6. Metabolite-Transcript Correlation Analysis

To elucidate the molecular mechanisms underlying metabolic changes across different safflower flowering stages, we conducted an integrative analysis combining transcriptomic and metabolomic data. Differentially expressed genes (DEGs) (FDR ≤ 0.01 and |log2FC| > 1) and significantly different metabolites (SDMs) (VIP > 1, P < 0.05, and |log2FC| > 1) were selected for correlation-based integration. The Pearson correlation coefficient, a widely used parametric method for assessing associations, was applied to identify significant correlations between DEGs and SDMs. To provide a comprehensive overview of the relationships between metabolites and transcripts in safflower, we present a correlation heatmap and network map (Figure 8C) highlighting key metabolite–transcript interaction modules.

### 4.7. qRT-PCR Analysis

Six flavonoid biosynthesis-related structural genes were validated by real-time fluorescence quantitative PCR (RT-qPCR). Total RNA from petals of three periods was isolated using the Trizol method. Reverse transcription was performed using the KangWei Century kit. Specific primers were designed in the laboratory and synthesized by Shanghai Bioengineering (Appendix A). The quantitative real-time PCR (qRT-PCR) reaction mixture consisted of 2 mL of template cDNA, 0.2 μM each of forward and reverse primers, and 2X MagucSYBR Mixture 10 mL. PCR amplification was performed using the QuantStudio3 real-time PCR System (Applied Biosystems, Thermo Fisher SCIENTIFIC, Shanghai, China) with an annealing temperature of 60 °C. Each analyzed single gene was tested with three biological replicates and three technical replicates. Quantitative data were analyzed using the 2^−ΔΔCt^ method with the 18s gene as an internal standard.

### 4.8. UV-B Stress Treatments

To investigate the impact of UV-B stress on flavonoid accumulation in safflower, we conducted a UV-B treatment experiment. Based on findings by Wang et al. [72], which demonstrated that UV-B irradiation enhances flavonoid levels in soybean calluses, we applied a similar approach to safflower petals. Blooming safflower plants were subjected to UV-B irradiation with a light intensity of 20,000 lx. Petals were collected at five time points: 0 h, 24 h, 48 h, 60 h, and 72 h post-treatment. At each time point, samples were harvested for analysis of total flavonoid content and for RNA extraction to assess gene expression changes. This experimental design aimed to elucidate the effects of UV-B stress on flavonoid biosynthesis and associated gene regulation in safflower.

## 5. Conclusions

In this study, comprehensive metabolomic and transcriptomic analyses were performed on the petals of safflower at the IN, FU, and FA stages. Because of the higher expression levels of genes related to phenylpropanoid and flavonoid biosynthesis pathways in petals during the FA, the content of flavonoids (especially kaempferol and quercetin derivatives) in petals during the FA was significantly higher than that in the other two periods. Through correlation analysis, DEGs, such as *FLS*, *DFR*, and *CHI*, and transcription factors, such as *MYB* and *WRKY*, whose expression levels were closely related to flavonoid concentrations, were identified, and these DEGs and TFs may be involved in flavonoid biosynthesis or translocation. Meanwhile, the differentially expressed genes all responded under UV-B treatment. In summary, this study provides a new perspective on the biosynthesis and accumulation mechanism of flavonoids in the safflower.

## Figures and Tables

**Figure 1 ijms-25-11903-f001:**
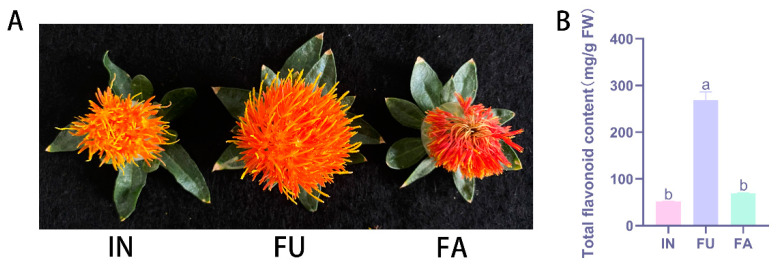
Phenotypic and TFC variation in three different flowering phases of safflower. (**A**) Phenotypic diagram of the developmental period of safflower. IN: initial flowering stage; FU: full flowering stage; FA: fade flowering stage. (**B**) Total flavonoid content in three periods (a,b: by ANOVA-two by two comparisons, the same letter above each column represents a non-significant difference, *p* > 0.05, and a different letter represents a significant difference, *p* < 0.05).

**Figure 2 ijms-25-11903-f002:**
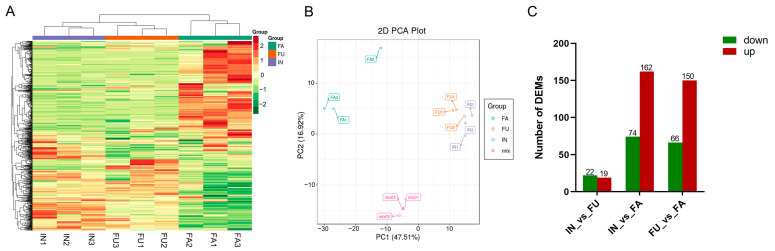
Differentially expressed metabolites in different flowering stages of safflower. (**A**) Heatmap of DEM in IN, FU, and FA flowering groups. (**B**) Principal component analysis of IN, FU, and FA flowering groups. (**C**) Statistics of DEM in IN, FU, and FA flowering groups.

**Figure 3 ijms-25-11903-f003:**
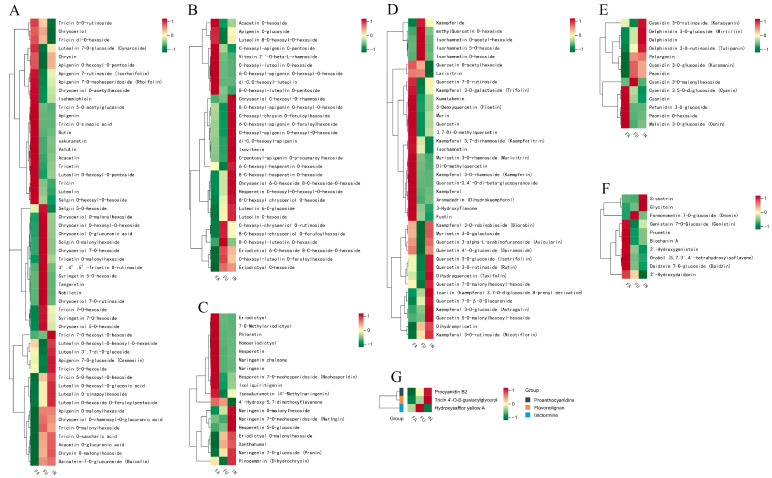
Heat map for cluster analysis of different subclasses of flavonoids: (**A**) flavonoids; (**B**) flavonoid carbonyl glycosides; (**C**) flavanones; (**D**) flavonols; (**E**) anthocyanins; (**F**) isoflavones; (**G**) flavone-lignan, chalcone analog, and proanthocyanidin. (Three replicates per group were used to construct a heat map using the average of the data from each group.)

**Figure 4 ijms-25-11903-f004:**
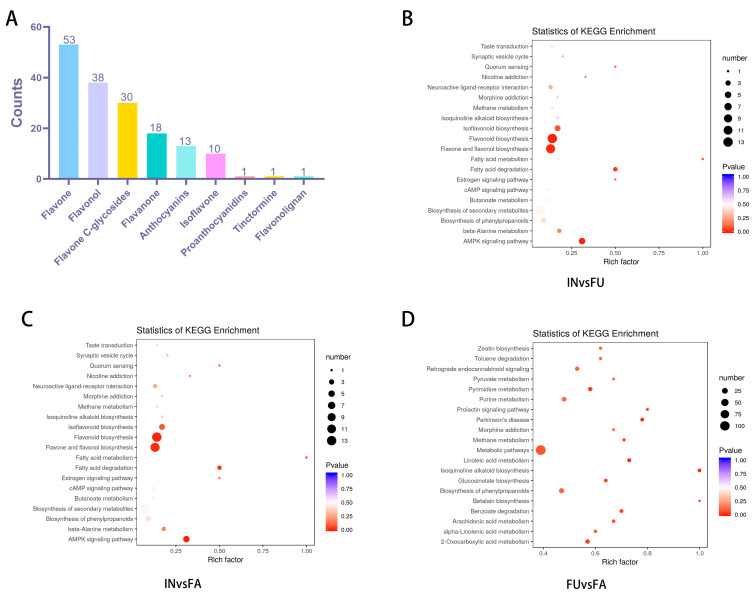
Functional annotation of DEM in different flowering stages of safflower. (**A**) The distribution and abundance of the differentially expressed flavonoid-typed metabolites identified among flowering stages. (**B**–**D**) KEGG pathway analysis of differentially expressed metabolite.

**Figure 5 ijms-25-11903-f005:**
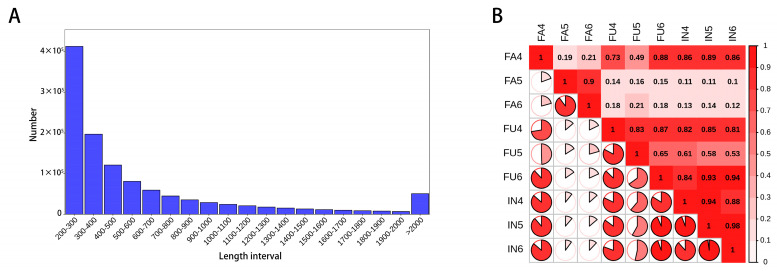
Genes annotation and correlation coefficients of intra- and inter-flowering group of safflower (**A**) Statistics of the unigenes distribution and lengths. (**B**) Heat map demonstrating the correlation coefficients of gene expression levels (FPKM) across intra- and inter-flowering groups.

**Figure 6 ijms-25-11903-f006:**
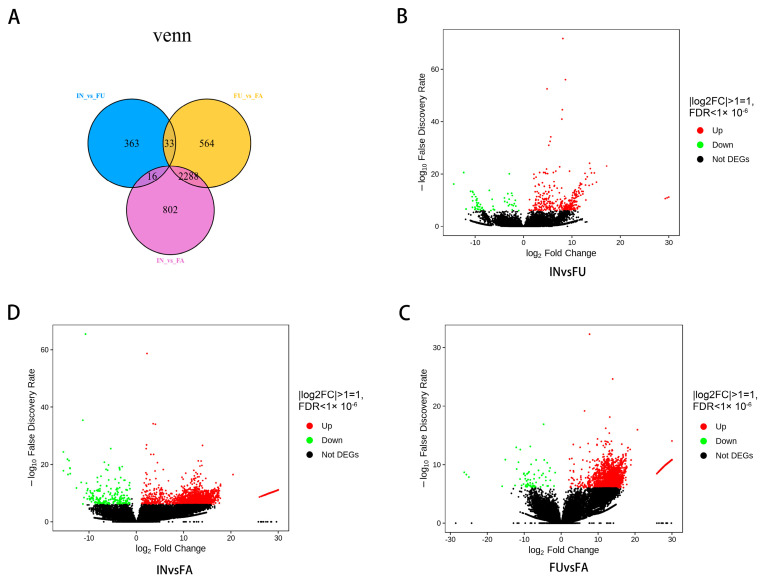
DEGs Identification between three different flowering stages. (**A**) The Venn diagram of differently expressed genes identified between IN, FU, and FA groups. (**B**–**D**) Visual illustration of the up and down-regulated DEGs between IN, FU, and FA groups using volcano plots.

**Figure 7 ijms-25-11903-f007:**
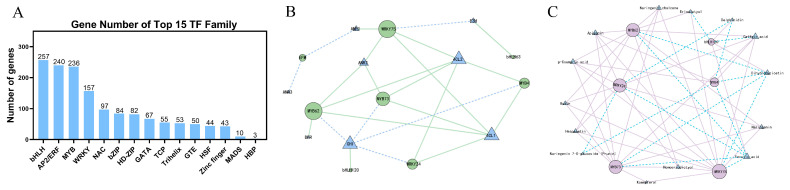
Correlation analysis of transcription factors with flavonoid key enzyme genes and flavonoids. (**A**) The top 15 transcription factor families with the highest expression in the three time periods. (**B**) Network diagram of the correlation between the eight TFs and the eight key structural genes for flavonoid biosynthesis. Green circles indicate TFs, blue triangles indicate structural genes for flavonoid synthesis, green solid lines indicate positive correlation, and blue dashed lines indicate negative correlation. (**C**) Correlation network diagram of 6 TFs with 14 flavonoid compounds. Purple circles indicate TFs, blue triangles indicate flavonoids, solid purple lines indicate positive correlations, and dashed blue lines indicate negative correlations.

**Figure 8 ijms-25-11903-f008:**
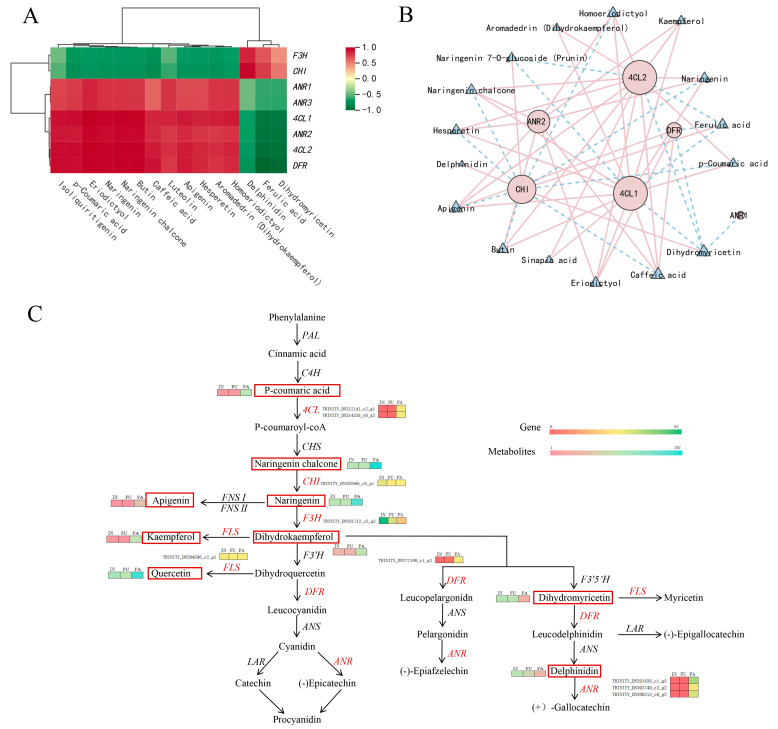
Correlation analysis and network map of differentially expressed genes (DEGs) and differentially expressed metabolites (DEMs) involved in the flavonoid and phenylalanine biosynthesis pathways. (**A**) Correlation heatmaps of the DEGs and DEMs. (**B**) Correlation network map of the DEGs and DEMs (|cor| > 0.8, *p* < 0.05; genes were depicted as circles and metabolites as triangles, with pink and blue lines representing positive and negative correlations). (**C**) Proposed biosynthetic pathway of flavonoid biosynthesis and reprogramming in different flowering stages of safflower based on the integrated analysis of metabolome and transcriptome profiling (genes in red font are DEGs and compounds in red boxes are DEMs).

**Figure 9 ijms-25-11903-f009:**
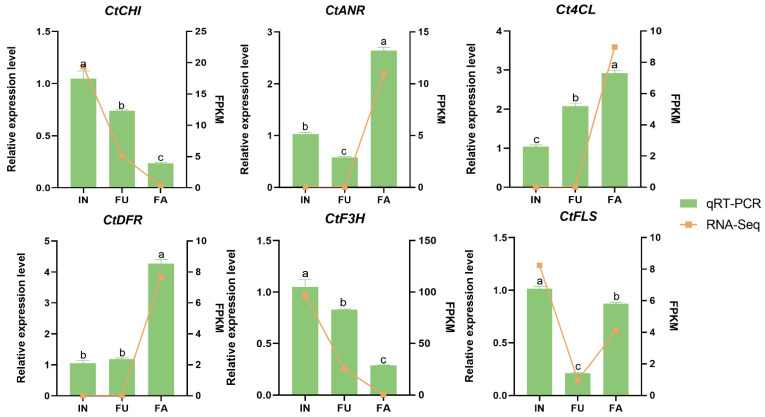
Gene expression validation of key functional genes involved in flavonoid biosynthesis in different flowering stages of safflower using qRT-PCR assay. The data were presented as means of three replicates, and error bars denote ± SE (n = 3), (a–c: by ANOVA-two by two comparisons, the same letter above each column represents a non-significant difference, *p* > 0.05, and a different letter represents a significant difference, *p* < 0.05).

**Figure 10 ijms-25-11903-f010:**
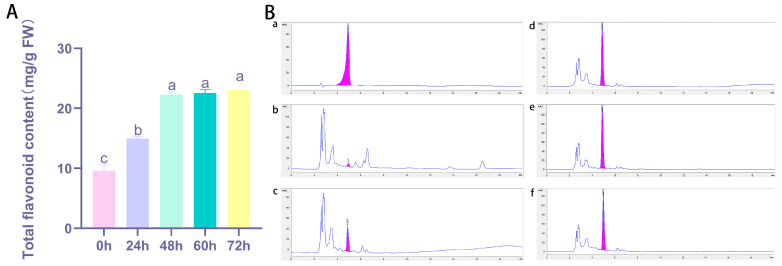
Quantification of total metabolite content in safflower under UV-B treatments. (**A**) Analysis of the total flavonoid content after UV-B treatment (a–c: by ANOVA-two by two comparisons, the same letter above each column represents a non-significant difference, *p* > 0.05, and a different letter represents a significant difference, *p* < 0.05). (**B**) Determination of total flavonoids by HPLC (**a**: 0.05 mg/mL Rutin standard), (**b**–**f**: UV-B-treated samples at 0 h, 24 h, 48 h, 60 h, and 72 h). Horizontal coordinate: time in min; vertical coordinate: response value in mAU.

**Figure 11 ijms-25-11903-f011:**
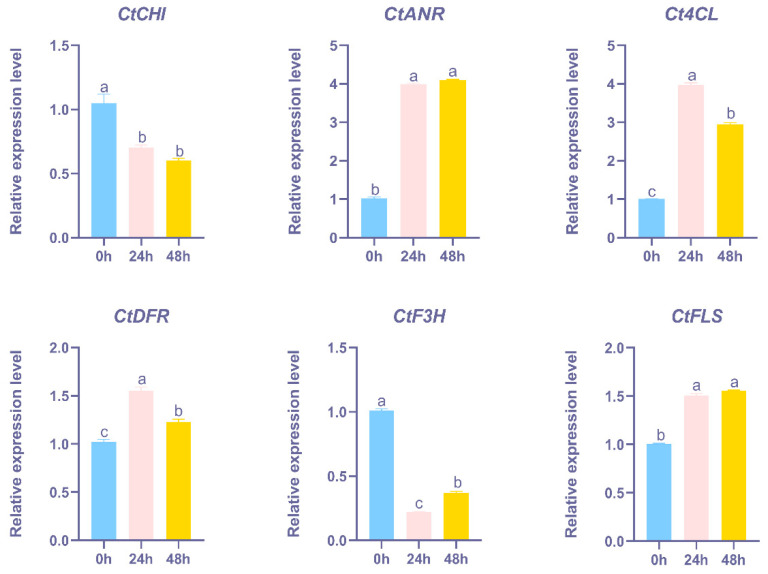
Relative expression level of key genes under UV-B treatments using qRT-PCR assay. The data are presented as means of three replicates, and error bars denote ± SE (n = 3), (a–c: by ANOVA-two by two comparisons, the same letter above each column represents a non-significant difference, *p* > 0.05, and a different letter represents a significant difference, *p* < 0.05).

**Table 1 ijms-25-11903-t001:** Pathway annotations of the core DEGS related to flavonoid biosynthesis in safflower.

Pathway	Gene ID	Symbol	Description
ko00940	TRINITY_DN312141_c2_g1	*4CL*	4-coumarate-CoA ligase
	TRINITY_DN314338_c0_g3		
ko00941	TRINITY_DN277190_c1_g3	*DFR*	flavanone 4-reductase
	TRINITY_DN281712_c2_g3	*F3H*	naringenin 3-dioxygenase
	TRINITY_DN294596_c2_g1	*FLS*	flavonol synthase
	TRINITY_DN302066_c8_g1	*CHI*	chalcone isomerase
	TRINITY_DN291838_c1_g5	*ANR*	anthocyanidin reductase
	TRINITY_DN302740_c2_g2		
	TRINITY_DN306215_c0_g2		

**Table 2 ijms-25-11903-t002:** Statistical table of correlation between DEGs and DEMs.

Gene ID	Positive Correlations	Negative Correlations
*4CL1* (TRINITY_DN312141_c2_g1)	pme2957, pme2319, pme0196, pme0379, pme3461, pme1436, pme1580, pme0376, pme0303, pme3473, pme1695, pme2963	pme2898, pme0305, pme0371
*4CL2* (TRINITY_DN314338_c0_g3)	pme2319, pme2963, pme1695, pme2957, pme0196, pme3461, pme0379, pme1580, pme0376, pme3473, pme0303	pme0305, pme0371, pme2898
*ANR1* (TRINITY_DN291838_c1_g5)	/	pme2898
*ANR2* (TRINITY_DN302740_c2_g2)	pme2319, pme2957, pme0196, pme3461, pme3473, pme0376, pme1580, pme0379, pme0303	/
*CHI* (TRINITY_DN302066_c8_g1)	pme0305, pme0442, pme2898	pme2957, pme0379, pme2319, pme1436, pme3461, pme0376, pme3473, pme0303
*DFR* (TRINITY_DN277190_c1_g3)	pme0379, pme1580, pme0303	pme2898, pme0305

## Data Availability

The sequencing data have been deposited into NCBI under the accession number PRJNA1160341. All data generated or analyzed during this study are included in the published article and Appendix A.

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
