# Peer review of "Integrated Metabolomics and Transcriptomics Provide Key Molecular Insights into Floral Stage-Driven Flavonoid Pathway in Safflower"

_ijms, 2024, doi:10.3390/ijms252211903_

Round 1
Reviewer 1 Report
Comments and Suggestions for Authors
In this article, Yu et al. combined with metabolomics and transcriptomics to insight into the molecular mechanisms of floral stage-driven flavonoid accumulation in safflower. The topic is interesting. However, some points need to be addressed and corrected before publication. All detailed comments are as follows:
1. Gene names should be in italics throughout the article.
2. In “Introduction” section, authors should add the relative references about flavonoids in safflower.
3. For metabonomic analysis and RNA-seq, authors should provide the correlation between the biological replicates of each sample.
4. For Figure 2A, it is suggested that all templates be put together for PCA.
5. In order to identify the key genes of flavonoid biosynthesis, it is suggested to perform the correlation analysis between metabolites accumulation and gene expression. In addition, the transcription factors should also be considered.
6. Figure 7, the FPKM value of genes should be added to the Figure to show the trend between qRT-PCR and RNA-seq.
7. For qRTPCR, why the 18s gene was selected as an internal standard? In my experience, the expression of this gene in different tissues is unstable.
8. Grammar check of this article are further needed by professionals.
9. Please check the structure of format presentation for International Journal Of Molecular Sciences.
Moderate editing of English language required.
Author Response
Reviewer#1:
In this article, Yu et al. combined with metabolomics and transcriptomics to insight into the molecular mechanisms of floral stage-driven flavonoid accumulation in safflower. The topic is interesting. However, some points need to be addressed and corrected before publication. All detailed comments are as follows:
- Gene names should be in italics throughout the article.
Response: Thank you very much for pointing this out. Based on your comments, we have italicized the gene.
- In “Introduction” section, authors should add the relative references about flavonoids in safflower.
Response: Thank you for your advice. The relevant references about flavonoids in safflower have been added in the revised version (Lines 39-42).
- For metabonomic analysis and RNA-seq, authors should provide the correlation between the biological replicates of each sample.
Response: Thank you for your valuable suggestion. We have added the correlation analysis of the RNA-seq and metabolomics analysis in figure 6 of the revised manuscript.
- For Figure 2A, it is suggested that all templates be put together for PCA.
Response: Thank you very much for pointing this out. We have modified figure 2A as suggested by worthy reviewer.
- In order to identify the key genes of flavonoid biosynthesis, it is suggested to perform the correlation analysis between metabolites accumulation and gene expression. In addition, the transcription factors should also be considered.
Response: Thank you for this insightful suggestion. We agree that correlating metabolite accumulation with gene expression is crucial for identifying key genes in the flavonoid biosynthetic pathway. In the revised manuscript, we have incorporated a detailed correlation analysis between flavonoid metabolite profiles and the expression levels of candidate biosynthetic genes at different flowering stages (Added figure 6). This allows us to pinpoint the most relevant genes involved in flavonoid biosynthesis in safflower during flowering development.
- Figure 7, the FPKM value of genes should be added to the Figure to show the trend between qRT-PCR and RNA-seq.
Response: Thank you for your valuable suggestion. We agree that including the FPKM values in Figure 7 would provide a clearer comparison between the qRT-PCR and RNA-seq results, thereby improving the visualization of gene expression trends. In the revised version, we have added the FPKM values for each gene alongside the qRT-PCR data, allowing for a direct comparison of expression levels obtained through both methods (now figure 8 in the revised version). This addition highlights the consistency between the RNA-seq and qRT-PCR data, further validating our findings. We have also updated the corresponding figure legend and description in the Results section to reflect this change.
- For qRTPCR, why the 18s gene was selected as an internal standard? In my experience, the expression of this gene in different tissues is unstable.
Response: Thank you for raising this important point. Based on previous experimental data and validation studies specific to safflower, we found that the 18s gene exhibits relatively stable expression across different tissues and developmental stages. We also conducted preliminary tests to confirm its stability under our experimental conditions before selecting it as the internal reference gene for qRT-PCR. However, we recognize that the expression stability of housekeeping genes can vary between species and experimental contexts. To address this, we will consider including a brief discussion in the revised manuscript acknowledging the potential variability in 18s gene expression and the steps we took to validate its use in our study.
- Grammar check of this article are further needed by professionals.
Response: Thank you for your feedback. We appreciate your suggestion and have re-checked the manuscript for a thorough grammar check by professionals before resubmission.
- Please check the structure of format presentation for International Journal of Molecular Sciences.
Response: Thank you for your valuable feedback. We have thoroughly reviewed the manuscript to ensure that it adheres to the formatting guidelines specified by the International Journal of Molecular Sciences.
Reviewer 2 Report
Comments and Suggestions for Authors
The authors present a rather comprehensive study using an integrated metabolomic and transcriptomic approach to investigate the changes in flavonoid content and synthesis of safflower petals. The results are clearly presented and conclusions are mostly backed by the data. Overall, the manuscript has the potential to be of high interest to the readers. However, certain sections, especially regarding the UV-treatment, need to be presented and discussed more comprehensively. The authors claim in line 326 that "These findings(..) clarify how UV-B stress influences flavonoid biosynthesis", but this is insufficiently supported by the results in the current state. Further, methods have been insufficiently described for the readers to be able to reproduce the experiments.
In the following are several points that need to be addressed by the authors to justify publication:
- 1. Introduction: It would be beneficial to include a more comprehensive section on the use and application of integrated metabolomics and transcriptomics approaches.
- Section 2.2: Very little information is given on metabolite identification. No information is given at all on the use of standard substances. The authors are firmly advised to stick to the guidelines of consortia such as the metabolomics standards initiative (e.g. doi.org/10.1186/2047-217X-2-13) and report their findings accordingly (i.e. metabolite identification level) in order for the reader to be able to contextualize the results. Further, it might be beneficial to include an additional table or figure describing the identified metabolites in more detail, e.g. grouped by compound class.
- Figure 2. Axis texts need to be bigger/better readable.
- Figure 3D. More context and information need to be given as to what is presented in this figure.
- Line 161-162: please rephrase
- Figure 4: There seems to be a very low correlation between the replicates FA4 and FA5/6. Can you elaborate as to why this is the case? This potentially needs to be addressed in the discussion section as well. Further, the plot lacks axis title and needs more explanation. What do the pie-charts represent?
- Section 2.6 The method on how the integration of transcriptomics and metabolomics data is missing. This is crucial as this the main peg of the manuscript and integration is not trivial (see e.g. doi.org/10.1093/bib/bbv090)
- Section 2.8: As mentioned above, the results of the gene expression analysis have to be described and discussed in more detail. What do the changes in the expression levels implicate? Is the metabolomics data of these samples available to substantiate the findings?
- Figure 8B: Axis titles are barely readable and need to be improved. Figure legend should more comprehensively describe the content.
- Section 4.2: As mentioned above, more information needs to be given on how the metabolites where identified. Have reference standards been used? Which transitions where monitored? This needs to be provided as supplementary table.
- Line 346: What extraction solution was used?
- Line 351: There is no 6500 ATRAP, do you mean QTRAP?
- Section 4.5: Was the same HPLC and column as in section 4.2 used? If yes please state this, if no please give more details on the equipment
- Supplementary Tables: Some columns have not been translated to English and are therefore of little use for the average reader, please translate.
Author Response
Reviewer#2:
The authors present a rather comprehensive study using an integrated metabolomic and transcriptomic approach to investigate the changes in flavonoid content and synthesis of safflower petals. The results are clearly presented and conclusions are mostly backed by the data. Overall, the manuscript has the potential to be of high interest to the readers. However, certain sections, especially regarding the UV-treatment, need to be presented and discussed more comprehensively. The authors claim in line 326 that "These findings(..) clarify how UV-B stress influences flavonoid biosynthesis", but this is insufficiently supported by the results in the current state. Further, methods have been insufficiently described for the readers to be able to reproduce the experiments.
Response: Thank you for your thorough review and constructive feedback. We appreciate your positive comments regarding the comprehensiveness of our study and the clarity of the results. In response to your concerns, we have thoroughly expanded the section discussing the UV treatment to provide a more comprehensive analysis of how UV-B stress influences flavonoid biosynthesis. We also incorporated additional data and relevant literature to strengthen our claims and ensure that the relationship between UV treatment and flavonoid synthesis is clearly articulated. In addition, we greatly acknowledge the need for a more detailed description of the methods used in our experiments. Following your suggestions, we have revised the Methods section to provide clearer, step-by-step protocols to ensure that our experiments can be accurately reproduced by readers. This includes specific details on sample preparation, treatment conditions, and analytical techniques employed.
In the following are several points that need to be addressed by the authors to justify publication:
- 1. Introduction: It would be beneficial to include a more comprehensive section on the use and application of integrated metabolomics and transcriptomics approaches.
Response: Thank you for your insightful suggestion. We agree that a more comprehensive introduction on the use and application of integrated metabolomics and transcriptomics approaches would provide valuable context for our study. Following your suggestion, we have added a paragraph in the introduction section of the revised manuscript (Lines 85-100).
- Section 2.2: Very little information is given on metabolite identification. No information is given at all on the use of standard substances. The authors are firmly advised to stick to the guidelines of consortia such as the metabolomics standards initiative (e.g. doi.org/10.1186/2047-217X-2-13) and report their findings accordingly (i.e. metabolite identification level) in order for the reader to be able to contextualize the results. Further, it might be beneficial to include an additional table or figure describing the identified metabolites in more detail, e.g. grouped by compound class.
Response: Thank you for your constructive feedback regarding the metabolite identification process described in Section 2.2. We acknowledge the need for a more detailed account of our methods and the importance of adhering to the guidelines set forth by the Metabolomics Standards Initiative. In the revised manuscript, we revised section 2.2 with the inclusion of these missing information on metabolite identification, which includes the detail of the standard substances used for metabolite identification and quantification, including their sources and the conditions under which they were utilized. In addition, the detailed table including grouped compounds identification in each group are given in Supplementary table 1.
-Figure 2. Axis texts need to be bigger/better readable.
Response: Thank you very much for pointing this out. We have modified figure 2 as suggested.
- Figure 3D. More context and information need to be given as to what is presented in this figure.
Response: Thank you for your valuable feedback regarding Figure 3D. We recognize the need for additional context to help readers better understand the significance of the data presented. In the revised manuscript, we provided a more comprehensive explanation in the figure legend and in the text regarding Figure 3D.
- Line 161-162: please rephrase
Response : Thank you for highlighting this point. We have rephrased these lines in the revised manuscript.
- Figure 4: There seems to be a very low correlation between the replicates FA4 and FA5/6. Can you elaborate as to why this is the case? This potentially needs to be addressed in the discussion section as well. Further, the plot lacks axis title and needs more explanation. What do the pie-charts represent?
Response : Thank you very much for your valuable comments. The large difference between FA4 and FA5/6 may be due to storage problems of FA4 during the delivery process or a small amount of degradation of the sample. For subsequent analysis of the data, the less discrepant FA5 and FA6 were used.
- Section 2.6 The method on how the integration of transcriptomics and metabolomics data is missing. This is crucial as this the main peg of the manuscript and integration is not trivial (see e.g. doi.org/10.1093/bib/bbv090)
Response : Thank you very much for pointing this out. We have now added a new method section (4.6) regarding the correlation analysis and integration of transcriptomics and metabolomics data used in this study. In addition, we also provided new figure (Figure 6) of these analysis in the results section.
- Section 2.8: As mentioned above, the results of the gene expression analysis have to be described and discussed in more detail. What do the changes in the expression levels implicate? Is the metabolomics data of these samples available to substantiate the findings?
Response :Thank you for very thoughtful comments. Following your comment, we have extended the discussion on results of the gene expression analysis in section 2.8.
-Figure 8B: Axis titles are barely readable and need to be improved. Figure legend should more comprehensively describe the content.
Response : Thank you very much for pointing this out. We have replaced the image with a higher resolution one and labeled the vertical and horizontal coordinates in the figure notes within the revised version.
-Section 4.2: As mentioned above, more information needs to be given on how the metabolites where identified. Have reference standards been used? Which transitions where monitored? This needs to be provided as supplementary table.
Response: Thank you for your insightful comment. We agree that providing more detailed information on the metabolite identification process is crucial for transparency and reproducibility. In response to your suggestions, we have revised section 4.2 with the following details:
- We have clearly indicated the reference standards that were used for the identification and quantification of the metabolites.
- We have described the specific mass spectrometry transitions that were monitored during the metabolomics analysis, providing details on the precursor and product ions used for each metabolite.
- To further clarify and enhance the presentation of these data, we included a supplementary table (Table S1) listing all identified metabolites.
- Line 346: What extraction solution was used?
Response: The extraction solution of Methanol-Water was used for extraction. It has been added in the method section within the revised manuscript.
- Line 351: There is no 6500 ATRAP, do you mean QTRAP?
Response: Yes, our apologies for this typo. It has been modified to QTRAP in the revised version.
- Section 4.5: Was the same HPLC and column as in section 4.2 used? If yes please state this, if no please give more details on the equipment
Response: Sorry for not making it clear in our previous draft. The HPLC and column in section 4.5 was different than the earlier. We have now added the details of the instrument model as well as the column specification within the revised manuscript.
- Supplementary Tables: Some columns have not been translated to English and are therefore of little use for the average reader, please translate.
Response : Thank you very much for pointing this out. The Chinese part of those tables has been translated into English.
Reviewer 3 Report
Comments and Suggestions for Authors
This study by Yu et al. investigated the molecular mechanisms underlying the dynamic changes in flavonoid biosynthesis and accumulation in safflower during flower development. It provides valuable results for optimizing the harvesting time of flowers for medicinal purposes and gene resources to improve flavonoid content in flowers. In addition, the authors revealed the effect of UV exposition on flavonoid content in flowers. This manuscript is suitable for publication in IJMS. However, some issues need to be addressed prior to the Editor’s decision.
1. I suggest modifying “integrated omics terrain” into “integrated metabolomics and transcriptomics” in the title.
2. Line 68. PAL is Phenylalanine ammonia-lyase.
3. Anthors identified in total, 53 flavonoids, 38 flavonols, 30 flavonoid carbonyl glycosides, 18 flavanones, 13 anthocyanins, 10 isoflavones, 1 flavone-lignan, 1 chalcone analog, and 1 proanthocyanidin. Construct heatmaps of each sub-class of flavonoids to facilitate the overview of their dynamic changes during flower development.
4. Line 249. pme0442 should be a metabolite.
5. Figure 6. It is recommended to use the metabolite names rather than IDs. The Same thing in the text. Also, you reported the gene IDs without mentioning their annotations.
6. Transcription factors (MYB, NAC, AP2/ERF, WRKY, etc.) are the key regulators of flavonoid biosynthesis in plants. Therefore, authors need to screen out key differentially expressed transcription factors.
7. The authors over-describe the results. It is suggested that the results be made clear and easily readable by deleting unnecessary sentences.
8. The discussion lacks mechanism insights into the observed variations. No in-depth analysis of the identified key DEGs, etc., was conducted.
9. Lack of information related to metabolite qualitative and quantitative identification, metabolite data analysis, and overall statistical analysis.
Comments on the Quality of English LanguageRevise mistakes throughout the manuscript.
Author Response
This study by Yu et al. investigated the molecular mechanisms underlying the dynamic changes in flavonoid biosynthesis and accumulation in safflower during flower development. It provides valuable results for optimizing the harvesting time of flowers for medicinal purposes and gene resources to improve flavonoid content in flowers. In addition, the authors revealed the effect of UV exposition on flavonoid content in flowers. This manuscript is suitable for publication in IJMS. However, some issues need to be addressed prior to the Editor’s decision.
Comment 1: I suggest modifying “integrated omics terrain” into “integrated metabolomics and transcriptomics” in the title.
Response : First of all, we are very grateful to reviewer 3 for taking his time in findings constructive comments that need attention. Following these comments and suggestions, we have revisited our manuscript and revised the points accordingly. The title has been updated to “integrated metabolomics and transcriptomics Provide Key Molecular Insights into Floral Stage-Driven Flavonoid Pathway in Safflower”
Comment 2: Line 68. PAL is Phenylalanine ammonia-lyase.
Response : We feel very sorry for our carelessness. It has been modified to Phenylalanine ammonia-lyase.
Comment 3: Anthors identified in total, 53 flavonoids, 38 flavonols, 30 flavonoid carbonyl glycosides, 18 flavanones, 13 anthocyanins, 10 isoflavones, 1 flavone-lignan, 1 chalcone analog, and 1 proanthocyanidin. Construct heatmaps of each sub-class of flavonoids to facilitate the overview of their dynamic changes during flower development.
Response : Thank you for your valuable suggestion. Following your comment, we have constructed the heat maps of each subclass (Figure 3) and included them in the revised manuscript.
Comment 4: Line 249. pme0442 should be a metabolite.
Response : Sorry for this lapse. It has been modified to metabolite in the revised manuscript.
Comment 5: Figure 6. It is recommended to use the metabolite names rather than IDs. The Same thing in the text. Also, you reported the gene IDs without mentioning their annotations.
Response : Thank you for your valuable suggestion. We have changed the IDs to the corresponding gene names and metabolite names in the revised manuscript.
Comment 6: Transcription factors (MYB, NAC, AP2/ERF, WRKY, etc.) are the key regulators of flavonoid biosynthesis in plants. Therefore, authors need to screen out key differentially expressed transcription factors.
Response : Thank you for your valuable suggestion. Following your suggestion, we have now added the analysis of transcription factors and included them in the subsection 2.6 of the revised manuscript.
Comment 7: The authors over-describe the results. It is suggested that the results be made clear and easily readable by deleting unnecessary sentences.
Response : Thank you for pointing this out. We have streamlined the description in the results section.
Comment 8: The discussion lacks mechanism insights into the observed variations. No in-depth analysis of the identified key DEGs, etc., was conducted.
Response : Thank you very much for pointing this out. The analysis of differential genes has been added to the discussion.
Comment 9: Lack of information related to metabolite qualitative and quantitative identification, metabolite data analysis, and overall statistical analysis.
Response : Thank you very much for pointing this out. Qualitative analysis of metabolites was derived from metabolic library data, which were constructed from standards (purchased from BioBioPha/Sigma-Aldrich). We have provided the details in Table S1 as well as in the the revised manuscript.
Round 2
Reviewer 2 Report
Comments and Suggestions for Authors
The authors have made the effort to address all raised questions and ad additional information where necessary. By doing so, the already high quality of the manuscript was significantly enhanced and crucially it is now possible to more easily understand and reproduce the methods.
The supplementary material section has not been updated yet, please do so. Other than that, I don't have any more concerns and deem the manuscript publishable as is.
Author Response
Comment: The authors have made the effort to address all raised questions and ad additional information where necessary. By doing so, the already high quality of the manuscript was significantly enhanced and crucially it is now possible to more easily understand and reproduce the methods.
The supplementary material section has not been updated yet, please do so. Other than that, I don't have any more concerns and deem the manuscript publishable as is.
Response : We are very thankful to reviewer 2 for his endorcements and appreciations. We are glad that our revised manuscript meet your expectations as well as the standards set by the journal. As for the supplementary materials, we have now updated it during this revision. Thank you very much for your efforts to improve the quality of work.
Reviewer 3 Report
Comments and Suggestions for Authors
The manuscript has been significantly improved, and all my concerns have been addressed. However, there are still some minor issues that need to be fixed before the Editor can make the final decision.
Figure 3. The HCA shows the variation in the relative contents of these metabolites between samples from the same developmental stage. This indicates potential quantification errors. To fix this, I suggest you to construct new heatmaps using the average values from each stage and avoid clustering the columns (samples). In addition, you can either use the name of the metabolites or mention where they can be found in the caption.
Line 420. Add some references to support MYB and WRKY regulation of flavonoid biosynthesis.
Comments on the Quality of English LanguageRevise minor mistakes
Author Response
Comments and Suggestions for Authors
The manuscript has been significantly improved, and all my concerns have been addressed. However, there are still some minor issues that need to be fixed before the Editor can make the final decision.
1. Figure 3. The HCA shows the variation in the relative contents of these metabolites between samples from the same developmental stage. This indicates potential quantification errors. To fix this, I suggest you to construct new heatmaps using the average values from each stage and avoid clustering the columns (samples). In addition, you can either use the name of the metabolites or mention where they can be found in the caption.
Response : Thank you for your comment. Following your suggestions, we have revised and re-constructed the heat maps in Fig.3 using the average values of the three sets of data as well as the compound names during the minor revisions.
2. Line 420. Add some references to support MYB and WRKY regulation of flavonoid biosynthesis.
Response : Thank you for your comments. We added relevant references related to WRKY and MYB transcription factors in line 420 of the revised manuscript. Changes were highlighted with green color font.
We are very thankful to reviewers and editors for helping us in improving this manuscript. we hope that after this minor corrections, it can be accepted for publication in this esteemed journal.